



# Impact of wind pattern and complex topography on snow microphysics during ICE-POP 2018

Kwonil Kim[1], Wonbae Bang[1], Eun-Chul Chang[2], Francisco J. Tapiador[3], Chia-Lun Tsai[1], Eunsil Jung[4], and Gyuwon Lee[1]

[1]Department of Astronomy and Atmospheric Sciences, Center for Atmospheric REmote sensing (CARE), Kyungpook National University, Daegu, Republic of Korea
[2]Department of Atmospheric Sciences, Kongju National University, Gongju, Republic of Korea
[3]Earth and Space Sciences Research Group, Institute of Environmental Sciences, University of Castilla-La Mancha, Spain
[4]Department of Advanced Science and Technology Convergence, Kyungpook National University, Sangju, Republic of Korea

**Correspondence:** Gyuwon Lee (gyuwon@knu.ac.kr)

**Abstract.** Snowfall in north–eastern part of South Korea is the result of complex snowfall mechanisms due to a highly–contrasting terrain combined with nearby warm waters and three synoptic pressure patterns. All these factors together create unique combinations, whose disentangling can provide new insights into the microphysics of snow in the planet. This study focuses on the impact of wind flow and topography on the microphysics drawing of twenty snowfall events during the ICE-POP 2018 (International Collaborative Experiment for Pyeongchang 2018 Olympic and Paralympic winter games) field campaign in the Gangwon region. The vertical structure of precipitation and size distribution characteristics are investigated with collocated MRR (Micro Rain Radar) and PARSIVEL (PARticle SIze VELocity) disdrometers installed across the mountain range. The results indicate that wind shear and embedded turbulence were the cause of the riming process dominating the mountainous region. As the strength of these processes weaken from the mountainous region to the coastal region, riming became less significant and gave way to aggregation. This study specifically analyzes the microphysical characteristics under three major synoptic patterns: air–sea interaction, cold low, and warm low. Air–sea interaction pattern is characterized by more frequent snowfall and vertically deeper precipitation systems in the windward side, resulting in significant aggregation in the coastal region, with riming featuring as a primary growth mechanism in both mountainous and coastal regions. The cold low pattern is characterized by a higher snowfall rate and vertically deep systems in mountainous region, with the precipitation system becoming shallower in the coastal region and strong turbulence being found in the layer below 2 km in the mountainous upstream region (linked with dominant aggregation). The warm low pattern features the deepest system: precipitation here is enhanced by the seeder–feeder mechanism with two different precipitation systems divided by the transition zone (easterly below and westerly above). Overall, it is found that strong shear and turbulence in the transition zone is a likely reason for the dominant riming process in mountainous region, with aggregation being important in both mountainous and coastal regions.





# 1 Introduction

Understanding the developing mechanism of heavy snowfall in the eastern part of the Korean Peninsula (Gangwon region) could have a great impact on improving the accuracy of forecasts in winter. The snowfall in this region is known to be characterized by strong orographic effects due to complex terrain in the west (Taebaek Mountains) and by the air–mass transformation in the nearby ocean in the east (East Sea). The orographic enhancement and sea effect induce heavy snowfall in this region
(Chung et al., 2004; Cheong et al., 2006; Cho and Kwon, 2012; Jung et al., 2012, 2014). The complexity of the snowfall mechanism arises from the steepness of terrain from the Taebaek mountains to the ocean (horizontal distance of about 20 km and vertical heights of 1000 − 2000 m) in combination with air–mass transformation over the warm ocean. Thus, complex airflows of this region affect microphysics, in particular the snow growth process above the melting layer.

The details of the interactions between airflows and microphysical processes have been partially investigated in the past.
Indeed, it is well recognized that the dynamics associated with the shear have important implications for two primary snow growth processes (i.e. aggregation and riming). Under strong shear conditions, the turbulent cell and induced updraft are favored, being responsible for the production of a considerable amount of supercooled water that promotes both aggregation and riming as they increase the probability of collision between hydrometeors, in particular, snowflakes and supercooled water droplets (Houze and Medina, 2005). Colle et al. (2014) also investigated the relationship between degree of riming and radar
measurables (Doppler radial velocity, spectrum width, etc) from vertically pointing radar and determined the degree of riming every 15 − 30 minutes for twelve cyclone events. They reported turbulent motions and strong updrafts and downdrafts in heavy riming events. Further insight into the role of the warm conveyor belt and its dynamics on the snow microphysics in the Gangwon region was provided by Gehring et al. (2020b), who argued that ascents in the warm conveyor belt contributed to persisting production of supercooled liquid water and that the process enhances the riming process during extreme snowfall
events.

The mean air flow governed by the synoptic patterns control snowfall in the region. Numerous studies have reported that favorable synoptic conditions for heavy snowfall are the eastern expansion of Siberian high pressure and a cyclone passing through the central or southern part of the Korean Peninsula (Lee and Kim, 2008; Park et al., 2009; Jung et al., 2012; Cho and Chang, 2017; Kim et al., 2019a). The eastern expansion of Siberian high pressure (the so-called 'Kaema high pressure') is
responsible for snowstorms by air–mass transformation, which is induced by cold easterly or northeasterly flow over the East Sea. Sea–effect snow (Estoque and Ninomiya, 1976; Nakamura and Asai, 1985; Ikeda et al., 2009; Kindap, 2010; Nam et al., 2014; Bao and Ren, 2018; Steenburgh and Nakai, 2020) is the same kind of mesoscale precipitation caused by the air–mass transformation of cold air over warm ocean. It develops over the East Sea and affects downstream of the wind flow. This mechanism is not greatly different to the lake–effect snow (Laird et al., 2009; Minder et al., 2015; Kristovich et al., 2017;
Wiley and Mercer, 2020), except in that the source of moisture and heat fluxes is here the ocean and not a lake.

It is also known that low–pressure systems passing through the southern part of the Korean Peninsula support the easterly or northeasterly flow over the East Sea in the later part of the low–pressure passage and supply abundant moisture from the Southern Ocean, leading to heavier snowfall (Seo and Jhun, 1991; Park et al., 2009; Gehring et al., 2020b). Song et al. (2016)





classified the synoptic environments when Siberian highs expand eastward with and without low–pressure systems. They

showed that the mean precipitation amount increased by about 45% in the presence of both Kaema high and low pressure.

The complex and steep orography of the Taebaek Mountain ranges is another important factor for both the dynamics and the precipitation system (amount, distribution, and duration). The orographic effect on airflow and precipitation is difficult to characterize because of the many different mechanisms, as many as twenty, following Houze (2012). Yu et al. (2007) also pointed out that multiple precipitation cells produced by different mechanisms can appear simultaneously in complex terrain.

Previous studies on the impact of the Taebaek mountains in the precipitation systems were carried out mainly by numerical simulation. Lee and Kim (2008) examined the effect of the Taebaek mountains on the distribution and intensity of snowfall by removing their topography in numerical experiments. They showed that, with the mountains, updraft is strengthened from $0.2 \, \mathrm{m\,s^{-1}}$ to $1.2 \, \mathrm{m\,s^{-1}}$ and the accumulated precipitation amount increases 8.5 times more in mountainous areas than the experiment without mountains. They also found that a topographic blocking by the Taebaek mountains causes horizontal

convergence and stronger updraft along the coastal region, resulting in heavy snowfall. Lee and Lee (2003) also investigated the effect of the Taebaek mountains by numerical simulation for two heavy snowfall events. They reported that the Froude number, which can represent the blocking degree, is important to determine whether the precipitation amount is more in the coastal region or in the mountainous region.

It is also known that the wind flow plays a critical role in the precipitation patterns of this region. In particular, easterly

flow or northeasterly (or sometimes northerly) flow is thought to be necessary for the development of heavy snowfall (Nam et al., 2014; Cho and Chang, 2017; Tsai et al., 2018; Kim et al., 2019a). The term Korea easterlies (Kor'easterlies hereafter) was proposed by Park and Park (2020) to emphasize the unique features of the easterly flow in the East Sea, which bring sufficient moisture and interact with topography. In relation to the Taebaek mountains, the Kor'easterlies can be interpreted as a cross–barrier flow. The observation–based study of the evolutions of wind flow and precipitation in this region was carried

out by Tsai et al. (2018). They used three–dimensional high–resolution wind field retrieved from six Doppler radars and found that a small temporary change of direction or speed of cross–barrier flow can considerably affect the distribution and intensity of precipitation. Kim et al. (2019b) discussed whether it is possible to determine precipitation over the Gangwon region only by synoptic pressure patterns, and found that the synoptic pattern (eastward expansion of Siberian high) itself is not the only factor determining precipitation. They suggested the wind speed of cross–barrier flow and thermal stability, which determine

the Froude number, should be examined together with pressure patterns. Another interesting characteristic in this region led by the Kor'easterlies is cold air damming in the eastern region of the mountain range (Bailey et al., 2003; Lee and Xue, 2013; Nam et al., 2020). As the Kor'easterlies are blocked and weakened by the mountain barrier, the flow favors their being turned into northwesterly along the barrier resulting from the weakened Coriolis force due to the disrupted balance between the Coriolis and pressure gradient force (Bailey et al., 2003; Lee and Xue, 2013). The air mass in the eastern region of the barrier then

becomes cold, stable and dense as cold air from the north is advected by the northwesterly and accumulated, which is referred to as cold air damming. This cold air damming creates a strong convergence zone, resulting the thermal contrast between the dammed air and East Sea being intensified and the maximum precipitation shifted away from the Taebaek Mountains (Jung et al., 2012; Lee and Xue, 2013). The updraft can also be induced by low-level cold air damming in coastal areas because





its density is relatively higher than that of the air mass coming from the ocean (Tsai et al., 2018). It is also worth noting that

when the layer with the Kor'easterlies is generated, we expect that a layer of obvious directional wind shear is located between the westerly flow aloft and the Kor'easterlies below. It implies that shear–generated turbulence may be present and contribute to some microphysical processes. In spite of these precedents, little research has been conducted in the area to examine the impact of the synoptic patterns and topography on the microphysical processes. One of the main reasons for this was the lack of systematic microphysical observations in this region (Nam et al., 2014). It is necessary to explore how the snow microphysical

characteristics differ according to both synoptic patterns and terrain to fully understand winter precipitation in the Gangwon region. In spite of great efforts in the previous studies, a comprehensive and intensive study had not been conducted before 2016 due to the limited availability of intensive microphysical observations.

A field campaign named ICE-POP 2018 (International Collaborative Experiment for Pyeongchang 2018 Olympic and Para-lympic winter games) took place in the Gangwon region to address this issue (Gehring et al., 2020a, b; Lim et al., 2020; Jeoung

et al., 2020). The ICE-POP 2018, contributed to by 29 agencies from 12 countries, was a World Weather Research Program (WWRP) of the World Meteorological Organization (WMO) and was led by the Korea Meteorological Administration (KMA). One of the main scientific goals of the field campaign was to understand the complex winter precipitation with intensive mi-crophysical observations. For two winter seasons (2016–2017 and 2017–2018 winters), a dense observational network was operated ranging from ground microphysical instruments to remote sensing. The 2017–2018 winter added 19 additional sites

operated as supersites. MRRs (Micro Rain Radar) and PARSIVELs (PARticle SIze VELocity) were located in selected loca-tions both along the coastal line and crossing the Taebaek mountains (parallel to the coastal line). Radiosondes were launched at high temporal resolution in more than 8 sites in the Gangwon region during the field campaign. A microphysical study in the Gangwon region with this kind of observational network had not previously been carried out. Such unique comprehensive observations and datasets created during ICE-POP 2018 now enable progress towards a deeper understanding of the complex

winter precipitation in the region.

The purpose of this paper is to elucidate the microphysical characteristics of snow in the Gangwon region in both different heights and surface by using MRR and PARSIVEL datasets obtained in the two winter seasons during the ICE-POP 2018 project. We analyze all the events that were jointly observed by both remote and in situ instrumentation. Such a combination makes the research unique in terms of the datasets and allows us to provide new results about the microphysics in the region.

The article is organized as follows. Section 2 describes how observations were performed during ICE-POP 2018, the sites and instruments used for analysis, and how data were processed and analyzed. In Section 3, the three main synoptic conditions which were responsible for the heavy snowfall during ICE-POP 2018 are presented. In Section 4, the general microphysical characteristics of snow and the characteristics for each synoptic pattern are presented. The conclusions of this study (Section 5) transcend the cases presented and provide insights into the microphysics of snow.





## 2 Data and Methodology

### 2.1 Experimental design

The ICE-POP 2018 field campaign employed nineteen supersites which include ground and remote sensing microphysical instruments, six fixed 3–hourly radiosounding sites (In et al., 2018), four mobile sounding sites including dropsonde from research aircraft (Jung et al., 2020) and radiosounding from mobile cars and the research ship Gisang–1 (Choi et al., 2020), and two wind profilers in both coastal and mountainous areas (see Fig. 1). The supersites are designed to be aligned both along the coastal line (from SCW to DHW site) and cross (from GWU to PCO site) the Taebaek Mountains. The PARSIVEL disdrometers had been deployed at all nineteen supersites as indicated as an open circle in Fig. 1b. The MRRs were collocated with the PARSIVEL disdrometers at ten supersites (filled circle). We take advantage of the site design to investigate the vertical structure and size distribution of snow according to the topography (crossing the Taebaek Mountains), which is characterized by steep orography that rises abruptly from the coast to an altitude of 800 m (see Fig. 1b).

We selected five sites for the study with the criteria: 1) being aligned in transect (dashed line in Fig. 1a) from a mountainous area to coastal area so that they capture the precipitation evolution and modification with the cross–barrier flow, 2) stable collection of data during the campaign, and 3) having a good environment for the observation (i.e. no taller trees or buildings nearby the MRR antenna and PARSIVEL). Based on the criteria, we selected YPO (YongPyong Observatory; 37.6433° N, 128.6705° E, 772 m MSL), MHS (MayHills Supersite; 37.6652° N, 128.6996° E, 789 m MSL), CPO (Cloud Physics Observatory; 37.6870° N, 128.7188° E, 855 m MSL), BKC (BoKwang1–ri Community center; 37.7382° N, 128.7586° E, 175 m MSL), and GWU (Gangneung–Wonju national University; 37.7717° N, 128.8703° E, 36 m MSL) from west to east. Three sites (YPO, MHS, and CPO) are located in mountainous regions (Yeongseo, Pyeongchang county) and the others (BKC and GWU) are in coastal regions (Yeongdong, Gangneung city).

During the field campaign, twenty events were observed by collocated MRR and PARSIVEL at five sites (Table 1). The MRR and PARSIVEL measurements of a total of 374 hours (15.6 days) were used to analyze the vertical structure and size distribution in the Gangwon region.

### 2.2 Instruments

This study analyzes measurements from MRR, which is the K-band (24.23 GHz) FM–CW vertical pointing radar. MRRs were deployed at 10 sites in total during the field campaign (Petersen et al., 2018; Gatlin and Wingo, 2019). The MRR can provide vertical profiles of the equivalent reflectivity factor ($Z_e$), mean Doppler velocity ($V_D$), and the Doppler spectral width ($SW$). These variables are used to gather detailed information on the vertical structure of precipitation according to the observational sites.

The range (vertical) resolution of MRR is adjustable with 31 range steps. During the field experiment, identical observational strategy was assigned to MRRs for each site. The time resolution was 10 seconds and the range resolutions were 200 and 150 meters for 2016–2017 and 2017–2018 winters, respectively.





Because this instrument was originally developed to observe liquid precipitation, it has been used for rain observation (Löffler-Mang et al., 1999; Yuter and Houze, 2003; Peters et al., 2005; Tokay et al., 2009; Adirosi et al., 2016). However, after Kneifel et al. (2011) and Maahn and Kollias (2012) showed that MRR can be utilized for snow observation, MRR started to

be actively used in snow studies (Kristovich et al., 2017; Souverijns et al., 2017; Durán-Alarcón et al., 2019; Vignon et al., 2019). We applied the post–processing algorithm proposed by Maahn and Kollias (2012), which is suited to snow (or weak rain) measurement. This algorithm improves the sensitivity of MRR by enhancing noise level estimation and de–alias Doppler velocity. Unless this algorithm is applied to the MRR measurement, upward moving particles will appear as very high Doppler velocity.

In addition to MRR, five laser–based optical disdrometers were utilized to see characteristics of size distribution on the ground. The PARSIVEL, manufactured by OTT Hydromet, provides the size and fall velocity of each hydrometeor at the ground (Löffler-Mang and Joss, 2000; Tapiador et al., 2010; Tokay et al., 2014). The measuring area of the instrument is 54 $\text{cm}^2$ ($180 \times 30$ mm), and minimum observable size and fall velocity are 0.2 mm and 0.2 m s$^{-1}$, respectively. The PARSIVELs used in the study were collocated with MRRs (Petersen et al., 2018; Petersen and Tokay, 2019), so that the size distribution of

snow was obtained at each MRR site.

The PARSIVEL has been widely used in microphysical studies for solid particles (Yuter et al., 2006; Aikins et al., 2016; Pokharel et al., 2017) as well as for rain (Friedrich et al., 2016; Park et al., 2017). As documented in Friedrich et al. (2016), however, PARSIVEL can suffer from splashing of particles (observed as a small diameter with large fall velocity when particles fall on the head of the sensor) and margin fallers (observed as a faster velocity than true fall velocity when particles fall

through the edge of the sampling area). Some limitations in measuring snow particles due to particle shape assumption are well described in Battaglia et al. (2010). To address these issues, we applied a quality control procedure that discards particles with a diameter of $< 1$ mm to avoid splashing and border effects (Yuter et al., 2006; Aikins et al., 2016). Further, particles with fall velocity of above 1.4 times of empirical fall velocity of rain (Atlas et al., 1973) were removed. To eliminate rain data, we classified the major precipitation type every minute based on the precipitation type classification mask of Yuter et al. (2006). If

the number of raindrops was greater than 80% of total number of particles, we considered the major precipitation type as rain and discarded this time interval.

### 2.3 Methodology

The vertical structure of snow can be characterized by CFAD (Contour Frequency by Altitude Diagram) analysis (Yuter and Houze, 1995) which represents the frequency of each variable as a function of height. Here, the frequency distribution is

normalized by the number of times. For the comparison between the sites, the CFADs are interpolated into a common height grid of 150 m since the height resolution of MRR is different between two winter seasons (200 and 150 meters). We can thus compare the values of CFADs at each height between sites, which is a similar approach to that of Minder et al. (2015) except for the height interpolation. The lowest two range gates are discarded in order to avoid a near–field effect (Maahn and Kollias, 2012; Minder et al., 2015). The CFADs are constructed from 1.5 km MSL height by considering the site height (maximum of

855 m) and the near–field effect.





The microphysical characteristics of snow at the ground level is analyzed by particle size distribution (PSD, N(D)) from PARSIVEL, which is calculated as

$$N(D_i) = \sum_j \frac{N_{i,j}}{A v_{i,j} \Delta t \Delta D_i},\tag{1}$$

where $i$ and $j$ are the index of the diameter and velocity channel, respectively, $D_i$ is the diameter of $i$-th diameter channel, $N_{i,j}$
is the number of observed particles of the $j$-th velocity bin in $i$-th diameter bin, $A$ is the horizontal measurement area (54 cm$^2$),
$v_{i,j}$ is the fall velocity of the $j$-th velocity bin in $i$-th diameter bin, $\Delta t$ is the sampling time interval, and $\Delta D_i$ is the width of $i$-th diameter channel.

Generalized characteristic number concentration ($N_0'$) and diameter ($D_m'$) are calculated with third and fourth moments as follows (Testud et al., 2001; Lee et al., 2004; Bang et al., 2020):

$$N_0' = \frac{M_3^5}{M_4^4},\tag{2}$$

$$D_m' = \frac{M_4}{M_3}, \text{and}\tag{3}$$

$$M_n = \sum_i N(D_i) D_i^n \Delta D_i,\tag{4}$$

where $M_n$ is $n$-th moment of particle size distribution in unit of m$^{-3}$ mm$^n$. The $N_0'$ and $D_m'$ can represent the shape of particle size distribution. Parameter $N_0'$ is proportional to the intercept parameter of an exponential PSD and $D_m'$ is the volume–
weighted mean diameter (Lee et al., 2004). We calculate generalized characteristic parameters every minute after the quality control process.

Snowfall rate ($SR$) is calculated from particle size distribution:

$$SR = \frac{\pi}{6\rho_w} \sum_i \frac{\rho(D_i) \sum_j N_{i,j} D_i^3}{A \Delta t},\tag{5}$$

with the assumed density–size relationship proposed by Brandes et al. (2007):

$$\rho(D) = 0.178 D^{-0.92},\tag{6}$$

where $\rho$ is the density of snow, and $\rho_w$ is the density of liquid water.

## 3   Classification by synoptic patterns

The synoptic patterns are classified into three categories in our study: "air–sea interaction", "warm low", and "cold low" (Jeoung et al., 2020). The following sections describe each synoptic pattern in detail and explain how observed events are classified
into three defined synoptic patterns.



## 3.1 Air–sea interaction

The "air–sea interaction" pattern is related to the air–mass transformation by cold air and relatively warm ocean. This is known to be related to the synoptic condition of the Siberian High pressure expanding eastward to Kaema Plateau (i.e. Kaema High pressure) located in the northeastern part of the Korean Peninsula. The snowfall mechanism is air–mass transformation by warm ocean surface and cold air outbreak. The precipitation system is initiated and developed by air–mass transformation over the East Sea, then propagates to inland areas due to Kor'easterlies, and is intensified as it is lifted by steep topography. The precipitation pattern is known to be linked with the cross–barrier flow (Cheong et al., 2006). The precipitation intensifies as the cross–barrier flow is strengthened, and dissipates as the cross–barrier flow is weakened when the pressure gradient is reduced or the wind direction changes. Because the precipitation cannot go further west beyond where the Kor'easterlies can reach it, the precipitation area is restricted in the eastern area of the Korean Peninsula. The Gangwon region can frequently have precipitation of this type. According to Choi and Kim (2010), the occurrence of the snowfall events in Korea associated with expansion of Siberian high is 43.8% for 36 years.

Previous studies have emphasized the importance of both synoptic and regional factors in the snowfall in this region (Nam et al., 2014; Kim et al., 2019a). It has been shown that the correlation between the precipitation in this region and sea level pressure over Kaema Plateau and Bering sea has intensified recently (after 1999) (Cho and Chang, 2017). Kwon et al. (2015) investigated low–level stratiform cloud associated with heavy snowfall and showed that it is mostly related to the expansion of Siberian High pressure with air–sea temperature difference of 7–12 °C. Nam et al. (2014) reported a correlation of $R$=0.67 between the snowfall amount in coastal areas and temperature difference between 850 hPa and sea surface temperature (SST) over the East Sea. Kim and Jin (2016) emphasized the importance of SST as a source of the energy for the formation of air–sea interaction snowfall. Their results suggest the warm SST anomalies in the East Sea result in heavy snowfall as the lower part of the air mass becomes warmer, which increases thermal instability, and moisture. The role of low–level instability by cold air outbreak and warm ocean in heavy snowfall were also highlighted by Lee et al. (2012). The strong surface heat fluxes produced by the East Sea are also known to be an important contributor to heavy snowfall in this region (Lee et al., 2018). Air–sea interaction pattern depends greatly on these factors.

Various terms have been used to refer to this synoptic pattern. It was referred to as air–mass transformation type (AT) in Cheong et al. (2006). It is also called east–coast terrain effect type (TE) (Cheong et al., 2006) since a snowstorm of this type is likely to interact with the complex topography. On the other hand, Song et al. (2016) called it 'cP high expansion' type or 'cP high expansion type + trough' type, emphasizing the cold and dry air mass. It is also referred to as stagnation type (Ko et al., 2016) as is often related to an atmospheric blocking pattern (Rex, 1950).

One of the examples of weather chart and sounding of this pattern is shown in Fig. 2. The radiosonde is launched in a coastal area. The northeasterly wind is induced in the East Sea and eastern coastal region along with the eastward expansion of the Siberian high to the East Sea (Fig. 2a). A saturated layer is observed from the surface to 790 hPa, along with the northeasterly or easterly winds. This layer is responsible for the snowstorm by air–sea interaction. The layer is bounded to the altitude where the Kor'easterlies exists. A strong thermal inversion layer is located from the height slightly higher than the upper boundary





of the saturated layer to 650 hPa with dry and weak wind speed of less than 5 m s$^{-1}$. The easterly winds in the lower level then turn to northwesterly and westerly above the inversion layer, with a dry layer up to 500 hPa. Note that the saturated layer above 500 hPa is not a typical situation in this pattern and was not related to the snowfall in this event.

### 3.2   Cold low

In the "Cold Low" pattern, the low pressure is located in the north of the polar jet stream and crosses the middle of the Korean
Peninsula. In this synoptic condition, the precipitation is associated only with a low–pressure system (not with induced airflow) and moves towards the east. The western area (mostly in the middle area) of the Korean Peninsula as well as the eastern area is expected to have precipitation as the low crosses the Korean Peninsula.

   The Gangwon region is not always expected to have heavy precipitation in this pattern. The moisture is consumed by precipitation in the western part of the mountains when the low system approaches the Gangwon region. The eastern coast
(leeward) usually has less precipitation than the mountainous area (windward) in the Gangwon region due to sublimation of snow by diabatic warming and drier conditions when crossing the mountains.

   Not only can precipitation occur when there is a well–developed low–pressure system, but also in the middle part of the Korean Peninsula even when a weak surface trough extends out over the Yellow Sea. In this case, the heated air parcel over the Yellow Sea, which forms warm advection under 850 hPa, is lifted by a cold pool inland, which is usually intensified in the
early morning. However, this system tends to precipitate mostly in the western part of the Korean Peninsula and is not likely to produce heavy snow in the Gangwon region. This type is considered as a cold low in this study.

   Since the low–pressure system is responsible for the precipitation, the snowfall rate depends on the location and intensity of the low–pressure system in contrast to air–sea interaction type. The Cold Low is the same type as the extratropical cyclone type (EC, Cheong et al., 2006) in which the precipitation is mainly produced by cold and warm fronts the extratropical cyclone
that crosses the middle part of the Korean Peninsula. This pattern is called the low crossing type in the synoptic classification by Ko et al. (2016). Song et al. (2016) interprets this pattern as a low passing type, but it is close to a low crossing type based on their description of this type where a low-pressure system crosses the middle part of the Korean Peninsula.

   Figure 2c shows an example of the surface chart of a cold low pattern. The low–pressure system was originated from Shandong Peninsula and propagated to the East Sea, leading to precipitation in the middle of the Korean Peninsula. It can be
seen from the vertical wind profile in Fig. 2d that the westerly is dominant except for near the surface. Wind speed in higher levels than 800 hPa is stronger than m s$^{-1}$. Saturated air is found up to 650 hPa, which is relatively higher than that of an example of air–sea interaction pattern (Fig. 2b).

### 3.3   Warm low

The "Warm Low" type indicates the synoptic situation where the low is located south of the polar jet stream and leads to
precipitation over the Korean Peninsula. The low is usually formed over eastern central China and moves through the south and southeast of the Korean Peninsula (see Fig. 2e). This synoptic type is the most complicated and has several interesting features that can only be found in the Gangwon region.





Tsai et al. (2018) showed that the system can be separated into two stages with different precipitation mechanisms during the passage of low–pressure system through the southern Korean Peninsula. As the low–pressure system approaches the Korean

Peninsula (early stage), precipitation that is directly related to low pressure affects the southern and central parts of the Korean Peninsula. A large amount of moisture from the warm ocean along the southwesterly flow is continuously supplied to the system, resulting in a vertically developed deep precipitation system. Tsai et al. (2018) demonstrated that the precipitation is associated with warm front and warm advection, which means the intensity of low pressure and snowfall rate are correlated. Additionally, the Kor'easterlies is induced in the eastern coast of Korean Peninsula by the low in the south, leading to air–mass

transformation in the East Sea. The Kor'easterlies and resulting air–mass transformation can be more intense if the Kaema High pressure system is present, being the synoptic condition of the air–sea interaction.

Thus, the system in the early stage is divided into two layers vertically. The precipitation (embedded in the low–pressure system) above the shear layer develops up to a high altitude of almost 10 km along with the westerly, and there is precipitation generated by Kor'easterlies below the shear layer (see Fig. 2f). In the layer where the westerly and Kor'easterlies meet, named

transition zone, there is a thermal inversion layer (see 670 hPa in Fig. 2f). The thermal inversion in the shear layer can also be found in Tsai et al. (2018) (see dash–dotted line and windbarbs in their Fig. 7) and Kim et al. (2018) (see their Fig. 6a) when the synoptic condition is similar to this stage. Kim et al. (2018) interpreted this vertical structure as a two–layer cloud and reported that the altitudes of both inversion and shear layers are at 3 km MSL. In contrast to the air–sea interaction pattern that is dry in the upper inversion layer, this pattern is mostly saturated, even in the upper layer.

In the later stage, when the low moves to the southeast of the peninsula, Kor'easterlies–associated precipitation remains, as the low pressure in the south can still induce an easterly flow in the East Sea. The Kor'easterlies tend to change from southeasterly to northeasterly because of eastward movement of low pressure (Tsai et al., 2018). The depth of precipitation is shallow and precipitation features are similar to those of air–sea interaction type.

The warm low type is thought to be a favorable condition for heavy snowfall. The north–south oriented high–to–low pressure

pattern is generally known to promote Kor'easterlies and heavy snowfall (Bae and Min, 2016; Lee et al., 2018), which is the case when Kaema high pressure is supported. When atmospheric blocking (Rex, 1950) restrains this system from propagating to the east, extreme snowfall events (snow depth of 192.6 cm within 9 days) in the Gangwon region could occur (Bae and Min, 2016). All the synoptic patterns of three extreme heavy snow events over 12 years (2001–2012) analyzed by Kwon et al. (2014) were similar to the warm low type. Nam et al. (2014) analyzed snowfall amount and frequency for 100 years (1912–2012) and

found that the Gangwon region has more frequent and more intense precipitation in late winter (i.e. February and March). This is consistent with the period when the polar jet moves north as the cold continental air mass is weakening, leading to a greater possibility of a warm low rather than a cold low.

The possible causes for the heavy snowfall could be as follows. 1) The substantial heat and moisture from both the south and east ocean of the Korean Peninsula can contribute to the precipitation system, 2) precipitation enhancement by a seeder–feeder

mechanism (Bergeron, 1965; Rutledge and Hobbs, 1983; Minder et al., 2015) is led by the microphysical interaction between falling seeding precipitation particles aloft generated by low–pressure system and the feeder orographic cloud below (note that the feeder cloud in this region is originally produced in the East Sea and is expected to be further developed by steep





orography), 3) thermodynamic instability in a low level over the East Sea cause the snow by air–mass transformation, resulting from cold and dry air brought by a cold–air outbreak over the warm ocean (Lee et al., 2012; Lee and Xue, 2013; Nam et al., 2014), 4) updraft occurs in the steep orographic ascent region (or coastal area due to orographic blocking or damming) by an orographic effect in unblocked condition, leading to production of supercooled liquid water and growth by riming (Medina and Houze Jr, 2003; Rotunno and Houze, 2007), and 5) both aggregation and riming processes are facilitated by a turbulent cell in a directional shear layer (Houze and Medina, 2005).

The distribution of accumulated precipitation is concentrated both in the southern region (directly influenced by precipitation embedded in the low–pressure system) and the east coast (indirectly produced by Kor'easterlies induced by low-pressure system), since this type has two stages and two layers. Thus, the precipitation area is sometimes used to determine this kind of synoptic pattern. Ko et al. (2016) referred to this type as low passing and they determined it by searching for precipitation events only in the southern Korean Peninsula and eastern coastal area. Here the term low passing may imply that the low passes through the southern part of the Korean Peninsula.

There have been several terminologies of this synoptic condition in previous studies. Song et al. (2016) called it the south trough type in their synoptic classification. Cheong et al. (2006) refers to the extratropical cyclone in the south of the Korean Peninsula type (ECS). They mentioned that the snowstorm is indirectly affected by extratropical cyclones and is supported by abundant moisture and orographic effect. They also reported that the relative importance of factors controlling the precipitation development could vary depending on the location of low pressure. In addition, when the expansion of Siberian high pressure and the low pressure passing through the southern sea occurs simultaneously, they used the term combined type (COM). Arguably, there will be more different terms depending on the perspective of looking at the pattern in the same synoptic condition, but most of these are consistent with the synoptic condition and precipitation features of our classification.

Figure 2e shows well–developed low pressure located in the southern part of the Korean Peninsula with a minimum mean sea level pressure of 993 hPa. Precipitation in the Gangwon region is in the early stage of a warm low pattern at this time. Easterly and northeasterly are shown in the eastern coast and East Sea since they are located north of low pressure. A strong directional wind shear and collocated thermal inversion is identified in Skew $T$-log$p$ chart (Fig. 2f). An obvious veering of wind just below and above the transition zone is found, indicating the warm advection. It is also revealed that the air is nearly saturated from the ground to the altitude over 300 hPa, indicating a vertically deep precipitation system. It is expected that temperature, moisture, and wind profile will change to be similar to the air–sea interaction type (Fig. 2b) in the later stage when the low–pressure system is located in the East Sea.

### 3.4 Classification of observed events

We have determined a synoptic pattern for each event based on our knowledge on each pattern. Basically, the weather map, composite radar image, and ground accumulated precipitation distribution provided by KMA were utilized. Precipitation distribution is obtained from a heated tipping rain gauge at automatic weather stations (AWS) of KMA. If Siberian high pressure expands to the east and precipitation is concentrated on the east coast, it was determined as an air–sea interaction pattern. If precipitation is concentrated in the western central region of the Korean Peninsula and the location of precipitation on the radar





image is directly related to the location of low pressure, it is classified as a cold low type. When the surface trough of low pressure extends to the Yellow Sea, snowfall due to warm advection in 850 or 925 hPa was also classified as a cold low type. It was determined as a warm low pattern when low pressure passed through the south of the Korean Peninsula and precipitation

was concentrated in the south and east coasts of the Korean Peninsula, and precipitation developed in the East Sea due to the indirect effect of low pressure as well as the Kor'easterlies.

It is necessary to confirm our classification using other available observation data. Radiosondes launched at GWW (Gang-Won regional Weather administration, 37.8046° N, 128.8554° E, 79 m MSL) near GWU on the east coast were utilized to check the vertical wind profile and depth of the precipitation system. We also checked soundings in the mountainous region

if necessary. Surface wind from AWS at GWW and DGW (DaeGwallyeong Weather office, 37.6773° N, 128.7188° E, 773 m MSL) near MHS in the mountainous region were also utilized to check the prevailing wind. In addition, we also checked wind profiles from UHF wind profilers at GWW and DGW. Satellite images of infrared and visible channels from Communication, Ocean, and Meteorological Satellite (COMS) were also utilized in order to check the location of clouds.

We chose twenty events when both MRR and PARSIVEL observed well for all sites (see Table 1). We identified five air–sea

interactions, five cold low, and ten warm low patterns among the events. Cheong et al. (2006) reported that the frequency of each synoptic pattern within 21 years (1981–2001) is 65, 14, and 30 for air–sea interaction (AT and TE), cold low (EC), and warm low (ECS and COM) type, respectively. The frequency ratio between cold low and warm low type is similar to the data set of Cheong et al. (2006) (the warm low type is almost double of cold low). However, air–sea interaction pattern in our studied events is relatively less frequent. The air–sea interaction type is mostly found in 2016–2017, while the cold low type occurred

only in 2017–2018. Note that the warm low events in 2017–2018 are mostly related to the heavy snowfall events during the ICE-POP 2018.

## 4 Results

### 4.1 General snow characteristics during ICE-POP 2018

In this section, the characteristics of vertical structure and snow size distribution from the ICE-POP 2018 period (i.e. 2017–

2018 winter, 12 events) are analyzed according to the topography to explore general characteristics of snow in the Gangwon region. During ICE-POP 2018, five sites (YPO, MHS, CPO, BKC, and GWU from the west to the east) were equipped with PARSIVEL and MRR. To examine the vertical structure of precipitation, CFADs of reflectivity, Doppler velocity, and spectral width are plotted for the five selected sites (Fig. 3).

The reflectivity is generally increased with decreasing heights at all sites, resulting from the growth of snow particles.

However, the increase can be characterized by different slopes of roughly 2.0 (0.5) $\mathrm{dBZ\,km^{-1}}$ above (below) 3.0 km height, particularly at CPO, BKC and GWU sites. The steady growth below 3 km heights is shown in the mountain sites, YPO and MHS. This may imply that some different microphysical processes occur based on the altitude of 3 km when comparing between coastal and mountainous regions. A distinctly different characteristic can also be seen in the CFAD of Doppler velocity. The values of frequency peak of Doppler velocity are $1 - 1.5 \mathrm{\,m\,s^{-1}}$ at all sites with a slight increase as height decreases,





consistent with typical terminal velocity of snow particles. However, an increase in Doppler velocity at 3.0 – 3.5 km clearly appears in mountainous (YPO, MHS, and CPO) sites, and weakly at BKC site. These increased Doppler velocities maintain their speed as height decreases. Since the Doppler velocity of the vertical pointing radar is the combination of reflectivity–weighted terminal velocity of particles and vertical wind, increases in Doppler velocity could be explained by faster falling particles or downdraft. We hypothesize that the increased Doppler velocity is mostly contributed to by faster falling particles.

This is because it is unlikely there was a sustained downdraft or updraft for most of the precipitation events in such a huge volume with a height of almost 2 km (1.5 – 3.5 km altitude).

It is also found that spectral width is noticeably increased at the level where the Doppler velocity is increasing (i.e. 3.0 – 3.5 km) at mountainous sites. The enhanced spectral width can be contributed to by either turbulence or diversity of fall speed of particles due to coexistence of non–rimed and rimed particles. The most likely causes for enhanced spectral width in 3.0 –

3.5 km is the turbulence, rather than the diversity of fall speed. If this is caused by diversity of fall speed, we should still see a similarly high value at the altitude below the layer, but we can see the spectral width is decreased as the height decreases. Note that the enhanced spectral width is stronger in mountainous sites and further intensified in western sites. The higher spectral width at 1.5 – 2.0 km height is also shown in mountainous sites, indicating the turbulence at this level because only the diversity of fall speed may not be attributable to such a high spectral width such as $1.0 \, \mathrm{m \, s^{-1}}$. In the coastal sites (BKC

and GWU), depositional growth or aggregation process is dominant and the degree of turbulence is lower because neither significant increased Doppler velocity nor spectral width is found.

Size distribution of snow on the ground is represented in generalized characteristic number concentration ($N_0'$) and diameter ($D_m'$) (Fig. 4). Normalized frequency of $N_0'$ and $D_m'$ for the YPO and GWU sites shows the mountainous site (YPO) has greater number concentration (mean value of $2.60 \, \log(\mathrm{m^{-3} \, mm^{-1})}$) and a larger diameter (mean value of $2.10 \, \mathrm{mm}$) compared

with the coastal site (GWU) having a mean number concentration of $1.53 \, \log(\mathrm{m^{-3} \, mm^{-1}})$ and a mean diameter of 1.89 mm, leading to higher snowfall rate at YPO. This is consistent with relatively higher normalized frequency of reflectivity in moderate intensity (0 – 15 dBZ) at YPO, although the reflectivity pattern is similar, with a mean value of approximately 12 dBZ at 1.5 km height for two sites.

To support the fact that YPO has heavier snow due to broader particle size distribution, the difference of normalized fre-

quency of $N_0'$–$D_m'$ between the two sites is shown in Fig. 5. The difference of normalized frequency of $N_0'$–$D_m'$ is consistent with that found in Fig. 3. Higher number concentration is more frequent for a given diameter, and a larger diameter for a given number concentration is found at YPO site, resulting in broader snow size distribution. This implies that higher snowfall rate is frequent and more active aggregation produces larger snowflakes at YPO.

## 4.2 Snow characteristics in air–sea interaction pattern

We divide the twenty observed events into three synoptic patterns, as described in Section 3. In this section, the snow characteristics during the air–sea interaction pattern are explored. Compared with the previous section, MHS and BKC sites are excluded because they were deployed during the 2017–2018 winter season only.





The echo top height is higher in the coastal region (GWU) compared with the mountainous region (YPO and CPO), indicating that the vertical depth of precipitation is decreased when the system crossing the Mountains (Fig. 6). The reflectivity is
rapidly increased until $3.0 \, \text{km}$ with increasing Doppler velocity as height decreases at all sites, suggesting depositional growth. Below $2.5 - 3.0 \, \text{km}$, primary peaks of reflectivity (larger reflectivity) show nearly constant values with steady increase of Doppler velocity. This implies that the riming is the main snow growth process in this layer regardless of site location. When riming occurs, Doppler velocity is increased because the mass of snow increases due to collection of supercooled liquid water, while the size of the snow does not greatly enlarge, resulting in nearly constant or smaller increase of reflectivity.

The Doppler velocity less than $0 \, \text{m s}^{-1}$ below $3 \, \text{km}$ are revealed at YPO and CPO, indicating that updraft exists in mountainous sites. These updrafts may be induced by uplift of airflow by topography. Updraft is also found at the GWU site with less frequency at higher level. In the layer where the updraft is present, there is a secondary peak of reflectivity and enhanced spectral width in mountainous sites up to a height of $3 \, \text{km}$. This may suggest that new ice crystals are generated at a low level due to adiabatic cooling resulting from the updraft in mountainous regions. Increased spectral width may be contributed to by
turbulence due to interaction between airflow and complex topography.

Although reflectivity is higher at CPO compared with GWU, the particles on the ground show higher frequency of big snowflakes at GWU, not CPO. The lower frequency of small snowflakes and higher frequency of big snowflakes at GWU indicate that the aggregation process is active in the coastal area. The aggregation signature can also be found in the histograms of characteristic diameter shown in Fig. 7. The mean values of characteristic diameter increase from west (YPO, $2.27 \, \text{mm}$) to
east (GWU, $3.79 \, \text{mm}$). The snowflakes with a diameter larger than $12 \, \text{mm}$ are shown only in GWU and broader distribution is found in GWU, which indicates strong aggregation is favorable in coastal regions. This result agrees with the findings of Bang et al. (2019), who conducted a similar analysis using two air–sea interaction events and showed the GWU site has greater number concentration and a larger diameter than the YPO site.

### 4.3 Snow characteristics in cold low pattern

In the cold low pattern, the moisture is likely consumed in the western part of the Korean Peninsula, resulting in less strong precipitation in the eastern part. As expected, the reflectivity generally showed a low mean value of less than $10 \, \text{dBZ}$, and the lowest mean value of $5 \, \text{dBZ}$ is shown in GWU at the bottom layer (Fig. 8). This can also be supported by the smaller values of both $N_0'$ and $D_m'$ at the GWU site, suggesting strong evidence of depletion or sublimation of snow. The CFAD value at the GWU site shows lower frequency, which means it snowed over less time, and echo top height is also decreased at the GWU
site. A possible explanation for depletion of snow in coastal region may be dry air mass in the coastal area and diabatic heating as the airflow descends through the steep mountain slope.

All sites are characterized by strong turbulence below $2 \, \text{km}$ MSL height, leading to high spectral width and high variability of Doppler velocity. The strong turbulence at this layer corresponds to that shown in the general characteristics (Fig. 3). The possible reason for the low–level turbulence may be the interaction between the complicated orography in the Taebaek
Mountains and the strong westerly. Note that the median wind speed of all cold low events was $\text{m s}^{-1}$ at DGW located between YPO and CPO, while they were 2.60 and $2.30 \, \text{m s}^{-1}$ for air–sea interaction and warm low events, respectively.





Interestingly, there were distinct differences between the two mountainous sites (YPO and CPO). Comparing two sites, although they have similar reflectivity and spectral width CFAD patterns, both aggregation and riming processes are favored at CPO as shown in high (low) frequency of large (small) particles on the ground and an increase in Doppler velocity ($1.5 - 2$ m s$^{-1}$) in Fig. 8e, respectively. Figure 9 illustrates the normalized frequency of characteristic diameter and number concentration at the ground as a function of snowfall rate. It is apparent from this figure that even though those two sites have the same snowfall rate, CPO is characterized by a lower number concentration with larger snowflake size for a given snowfall rate. This means the aggregation process plays an important role in the region between YPO and CPO sites even though the distance is 6.46 km only.

## 4.4 Snow characteristics in warm low pattern

As explained in Section 3, a warm low pattern is likely to have unique characteristics as a result of its unique precipitation mechanism and complex geographical features in the Gangwon region. The vertical depth of precipitation is at least higher than 4.5 km (Fig. 10), which is consistent with saturated air over 300 hPa shown in Fig. 2f. In contrast to the differential growth rate of reflectivity below and above 3 km found in general characteristics (Fig. 3), almost constant increases in reflectivity with decreasing height are identified at all sites.

A strong riming signature at mountainous sites is identified around 3 km height, which is also found in the general characteristics (Fig. 3). It can be seen that Doppler velocity obviously increases in the enhanced spectral width layer in $3.0 - 3.5$ km. As higher spectral width is found at YPO, the Doppler velocity is also found at higher values compared with CPO (see the Doppler velocities close to 2.5 m s$^{-1}$ in Fig. 10d). Both the frequency of Doppler velocity above m s$^{-1}$ and primary frequency peak of Doppler velocity of an approximately 1.0 m s$^{-1}$ rise in that layer ($3.0 - 3.5$ km) and maintain their velocity as they fall. A possible reason for this is a strong riming process occurs in the enhanced spectral width layer, leading to production of fast–falling particles.

The enhanced spectral width may be explained by directional shear. There is a strong directional wind shear layer between the easterly or northeasterly in the lower and westerly in the upper layer in the early stage of a warm low pattern, resulting in a turbulent layer. The transition zone where strong wind shear exists is generally with thermal inversion. Lee (1999) reported that the inversion layer is usually located at 700 hPa in the synoptic situation when the snowstorm is affected by the strength of a low–pressure system in the southeast of the Korean Peninsula. This altitude is consistent with an enhanced spectral width layer in Fig. 10. Thus, the layer of enhanced spectral width in $3.0 - 3.5$ km suggests that it is caused by wind shear. It is also found that the general altitude of the wind shear in this region is located between $3.0 - 3.5$ km MSL (see Fig. 2f). The shear–generated turbulence could produce the updraft cell which is responsible for generation of supercooled liquid droplets and favors both aggregation and riming processes (Houze and Medina, 2005; Kumjian et al., 2014; Grazioli et al., 2015). This is a probable explanation of why the riming signature is identified. It is also found that the further west, the higher is the spectral width found below 2 km, which is also found in the general characteristics in Section 4.1. This may be caused by mechanical turbulence as Kor'easterlies flow over the complex orography.





As discussed above, snow in the upper and lower layers was initiated and developed from different systems, respectively. The upper snow is generated by a low–pressure system in synoptic scale, while the lower snow is produced by interaction between cold air and warm ocean in mesoscale and orographic lifting. Snow crystals generated by large–scale forcing (acting as a seeder cloud) collects supercooled liquid water or collides with snowflakes below and becomes rimed or aggregated snowflakes as it falls through feeder clouds produced by mesoscale forcing. A possible evidence of snowfall enhancement by seeder–feeder

mechanism (Bergeron, 1965) may be found in the continuous increase in reflectivity of the primary peak (see normalized frequency > 2.5%). Comparing mountainous regions (YPO and CPO) and coastal regions (GWU), we can see that the increase of reflectivity at the GWU site below 3 km is more rapid (approximately $2.0 \, \mathrm{dBZ \, km^{-1}}$ for normalized frequency > 2.5%) than at mountainous sites (approximately $1.3 \, \mathrm{dBZ \, km^{-1}}$ for normalized frequency > 2.5%). There are two probable explanations for this result. First, snowstorm at the low level in coastal areas, which is produced by air–sea interaction type, is likely to

have more sufficient moisture than mountainous areas since it is closer to ocean. Second, aggregation is the primary growth mechanism in the feeder zone at the GWU site because the Doppler velocity does not show a significant increase (almost constant at $1.2 \, \mathrm{m \, s^{-1}}$). A large snowflake grown by the aggregation process can promote an increase in reflectivity that is approximately proportional to the fourth moment of particle size distribution in the Rayleigh scattering regime (Bukovčić et al., 2018, 2020; Matrosov, 2020). The low frequency of high number concentration above $500 \, \mathrm{m^{-3} \, mm^{-1}}$ for a small diameter (<

3 mm) in Fig. 10l confirms that small snowflakes are mostly consumed by aggregation. At YPO and CPO, on the other hand, Doppler velocity apparently increases in the layer below 3.5 km and reflectivity shows less enhancement, suggesting snow is likely to predominantly grow by riming processes in the feeder zone with a modest contribution of aggregation. Large diameter (> 5 mm) snowflakes found at YPO and CPO sites (Fig. 10) implies that the aggregation process is also an important growth mechanism in mountainous regions.

It is also worth noting that the reflectivity at the low level is the highest at CPO. It is possible to hypothesize that the seeder–feeder effect would have been strongest at CPO because both seeder and feeder precipitation could be strong, since the CPO is still not only close to the East Sea but also located in the mountainous site; thus it can be considerably influenced by embedded precipitation in the low–pressure system. The higher frequency of large diameter (> 5 mm) snowflakes at CPO compared to that of YPO (Fig. 10) may be explained by a stronger seeder–feeder effect at CPO.

Figure 11 depicts the frequency comparison of number concentration and diameter between YPO and GWU sites to compare between mountainous and coastal regions. The mean values of number concentration and diameter of YPO (GWU) are 2.02 (1.47) $\log(\mathrm{m^{-3} \, mm^{-1}})$ and 2.30 (1.75) mm, respectively, indicating the mountainous region is characterized by heavier snowfall with more and larger snowflakes. The size distributions are not much different to those of the general characteristics (Fig. 4) because the number of warm low events account for a half of all events.

## 510   5   Conclusions

The main purpose of the study was to demonstrate the impact of wind pattern and topography on microphysical characteristics in the Gangwon region. Twenty snowfall events during the ICE-POP 2018 field campaign for two years were examined with





collocated MRR and PARSIVEL data at the five sites aligned in crossing mountains. This study analyzed CFADs of equivalent reflectivity, Doppler velocity, and spectral width from MRR and distribution of generalized characteristic diameter ($D'_m$) and

number concentration ($N'_0$) of snow size distribution from PARSIVEL.

The twenty–event composite analysis suggests that snow at mountainous sites at YPO, MHS, and CPO tends to grow by riming with moderate aggregation, whereas snow growth in coastal sites at BKC and GWU is dominant by aggregation or vapor deposition. Signatures of riming are found as a notable higher frequency of enhanced Doppler velocity below $3.0 - 3.5$ km altitude. We hypothesize that the reasons for this is that the turbulent layer at $3.0 - 3.5$ km altitude, which can be seen by

higher spectral width, produces a large amount of supercooled liquid water and subsequently fosters riming, as suggested by Houze and Medina (2005). It is also revealed that the snowfall rate is more intense in mountainous sites resulting from broader size distribution.

Since the wind pattern is governed by the synoptic condition, this study divided the events according to the synoptic pattern in order to further discover possible linkages between airflow, topography, and characteristics. Synoptic patterns are classified into

three categories (air–sea interaction, cold low, and warm low). MRR observations for air–sea interaction suggests that riming exists at all sites. This may be explained by low–level updraft induced by orographic lifting and convective instability over ocean due to the Kor'easterlies. Updraft is more frequent in mountainous sites, leading to a stronger signature of riming. Size distribution characteristics from PARSIVEL indicate larger snowflakes by aggregation are more frequent in coastal regions. This is probably due to the moisture–rich condition because of the shorter distance from the ocean, which can also be as a

deeper depth of precipitation.

This study shows that, in a cold low pattern, precipitation in coastal regions can be characterized by relatively shallower, less intensive, and less turbulent snow with a smaller size and less number concentration, indicating strong sublimation of snow in the downwind side of the mountain ranges. The possible explanations for this are the diabatic heating due to descent of airflow and the relatively drier condition because moisture is mostly consumed in the west. The CFADs of spectral width suggest

that strong turbulence is located at low level at the mountainous sites, probably resulting from the interaction between complex terrain and high wind speed. It is speculated that turbulence plays a vital role in promoting aggregation in mountainous regions. This hypothesis is supported by the fact that the CPO site is characterized by greater aggregation for a given snowfall rate, in contrast to YPO.

The evidence that the wind pattern and topography affect microphysical processes can be clearly seen in the warm low

pattern. It is found that a layer with enhanced spectral width in $3.0 - 3.5$ km MSL is responsible for strong and obvious riming signatures beneath the layer. Directional wind shear generally located in $3.0 - 3.5$ km generates shear–induced turbulence, leading to intense riming, particularly in mountainous regions, by producing supercooled liquid water and increasing collision probability. MRR measurements suggest that the effect of the seeder–feeder mechanism on microphysics is probably different in mountainous and coastal regions. The mountainous region is more likely to present intensive riming with moderate aggre-

gation whereas the coastal region is likely to be affected by greater aggregation in the feeder zone. Reflectivity from MRR and size distribution from PARSIVEL suggest that seeder–feeder enhancement is thought to be the strongest at CPO. In addition, the size distribution of YPO and GWU indicates that a mountainous region has a heavier snowfall rate as seen in larger size



and larger number concentration. Given the physical mechanisms invoked to explain the observations these results are not necessarily particular of the region but likely applicable to other regions of the planet.

The present study provides evidence of the role of airflow and topography in microphysics in the Gangwon region by means of a comprehensive network of MRR and PARSIVEL. While this study is more focused on multi–storm characteristics, detailed analysis for each pattern or for a specific event will need to be undertaken to support the hypotheses and to answer questions remaining, such as the reasons for the stronger turbulence in the transition zone in the west in a warm low pattern. In future research, the use of observation–based three–dimensional wind or dual–polarization radar variables in the direction of

cross–barrier flow is suggested, as well as high–resolution simulation from numerical models.

*Data availability.*   Data are available upon request (Kwonil Kim via kwonil.kim.0@gmail.com)

*Author contributions.*   KK designed and performed the research under the supervision of GL. WB processed the PARSIVEL data. GL, EC, FJT, CT and EJ contributed to the scientific discussions and gave constructive advice. KK and WB carried out the MRR and PARSIVEL measurements. KK wrote the manuscript with substantial contributions from all co–authors. GL, FJT, and EC revised the manuscript.

*Competing interests.*   The authors declare that they have no conflict of interest.

*Acknowledgements.*   This work was funded by the Korea Meteorological Administration Research and Development Program under Grant KMI2018-06810. FJT acknowledges project PID2019-108470RB-C21. The authors greatly appreciate the participants in the World Weather Research Programme Research Development Project and Forecast Demonstration Project, International Collaborative Experiments for Pyeongchang 2018 Olympic and Paralympic winter games (ICE-POP 2018) hosted by Korea Meteorological Administration (KMA). We
would like to thank Sang-Won Joo, Yong-Hee Lee, Kwang-Deuk Ahn, Namwon Kim, and Seung-bo Choi at KMA for their support for the ICE-POP 2018 field campaign. The authors are grateful to Walter Petersen, Ali Tokay, Patrick Gatlin, and Matthew Wingo at NASA for providing the MRR and PARSIVEL instruments and processing the PARSIVEL data. We also would like to thank Byung-Gon Kim at Gangneung-Wonju National University and Byung-Chul Choi at High Impact Weather Research Center of the KMA for sharing their instruments. Finally, we thank Woo-Yeol Choi, Su-jeong Cho (currently at KMA), Choeng-lyong Lee, Daejin Yeom, Kyuhee Shin, DaeHyung Lee,
Eunbi Jeong, Geunsu Lyu, Hong-Mok Park, SeungWoo Baek, Heesang Yoo (currently at IBM), Youn Choi (currently at KMA), Bo-Young Ye, and Soohyun Kwon (currently at KMA) at Kyungpook National University for their great efforts to operate the instruments during the experiments over two years.





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



**Table 1.** Summary of events used in this study. The time is in UTC.

| No. | Year | Synoptic pattern | Date and Time [UTC] |
|---|---|---|---|
| 1 | | | Jan 04 2017 20:00 – Jan 05 2017 13:00 |
| 2 | | Air–sea interaction | Jan 30 2017 02:00 – Jan 30 2017 10:00 |
| 3 | | | Mar 02 2017 00:00 – Mar 02 2017 09:00 |
| 4 | 2016–2017 | | Mar 14 2017 00:00 – Mar 14 2017 10:00 |
| 5 | | | Jan 08 2017 00:00 – Jan 08 2017 20:00 |
| 6 | | Warm Low | Jan 29 2017 03:00 – Jan 29 2017 24:00 |
| 7 | | | Feb 21 2017 20:00 – Feb 22 2017 15:00 |
| 8 | | | Mar 01 2017 00:00 – Mar 01 2017 24:00 |
| 9 | | Air–sea interaction | Mar 15 2018 18:00 – Mar 16 2018 09:00 |
| 10 | | | Dec 09 2017 21:00 – Dec 10 2017 16:00 |
| 11 | | | Jan 07 2018 20:00 – Jan 08 2018 20:00 |
| 12 | | Cold Low | Jan 22 2018 03:00 – Jan 22 2018 22:00 |
| 13 | | | Jan 30 2018 07:00 – Jan 30 2018 24:00 |
| 14 | 2017–2018 | | Feb 22 2018 03:00 – Feb 22 2018 24:00 |
| 15 | | | Dec 23 2017 18:00 – Dec 24 2017 16:00 |
| 16 | | | Jan 16 2018 10:00 – Jan 17 2018 02:00 |
| 17 | | Warm Low | Feb 28 2018 00:00 – Feb 28 2018 24:00 |
| 18 | | | Mar 04 2018 15:00 – Mar 05 2018 09:00 |
| 19 | | | Mar 07 2018 05:00 – Mar 09 2018 04:00 |
| 20 | | | Mar 20 2018 18:00 – Mar 21 2018 14:00 |



**Figure 1.** (a) Locations of supersites and sounding sites of ICE-POP 2018, and (b) the vertical cross–section along thick dashed line in (a). Terrain height is filled in gray in (b). Five sites used in this study are colored red. The period of MRR and PARSIVEL measurements is noted with arrows.

**Figure 2.** (left) Surface weather chart from KMA for (a) air–sea interaction, (c) cold low, and (e) warm low type. (right) The radiosonde data plotted on a skew $T$-log$p$ chart in coastal area (GWW, near GWU) for (b) air–sea interaction, (d) cold low, and (f) warm low type. Skew $T$-log$p$ chart was created using the MetPy package (May et al., 2008 - 2021) in Python.



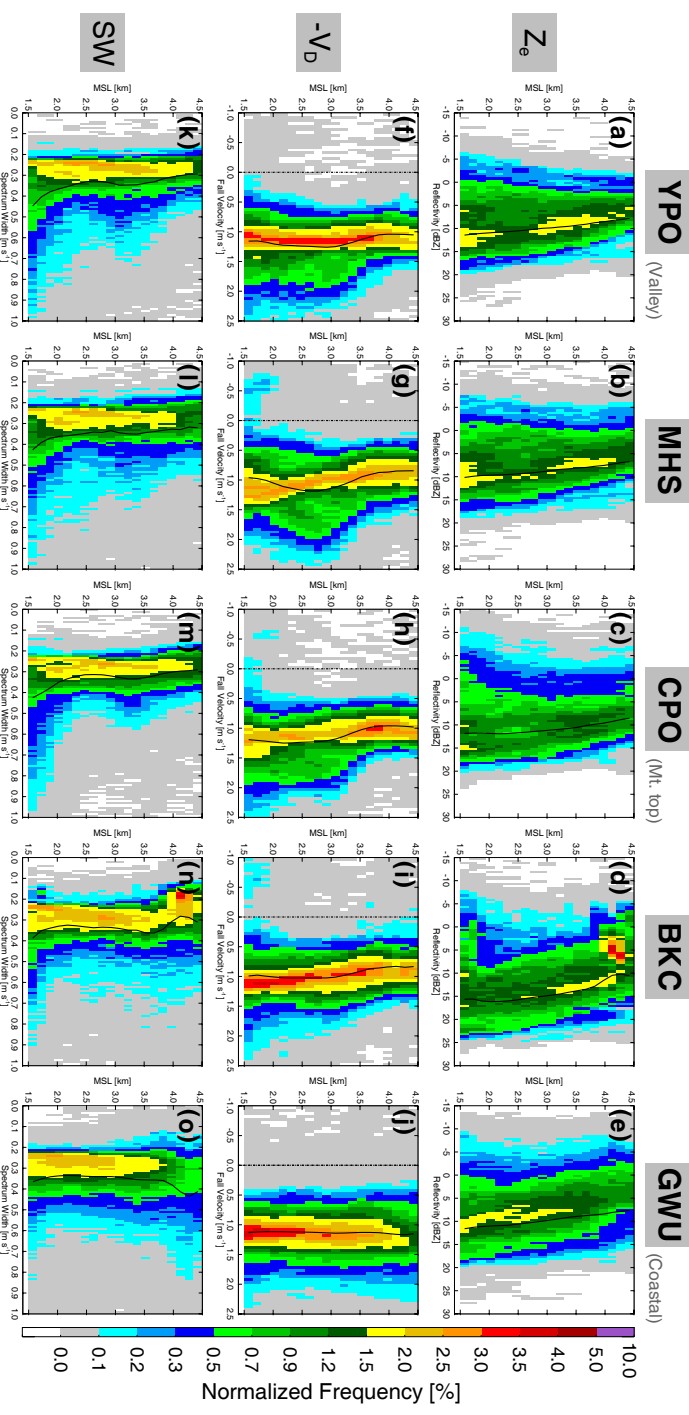

**Figure 3.** CFADs of radar reflectivity (dBZ, top), negative of Doppler radial velocity (m s$^{-1}$, middle), and spectral width (m s$^{-1}$, bottom) from MRR for 2017-2018 winter at (a, f, k) YPO at the valley, (b, g, l) MHS, (c, h, m) CPO at the mountain top, (d, i, n) BKC, and (e, j, o) GWU sites at the coastal region. Solid line indicates the average value for each diagram. Dash-dot lines represent 0.0 m s$^{-1}$ in Doppler velocity.

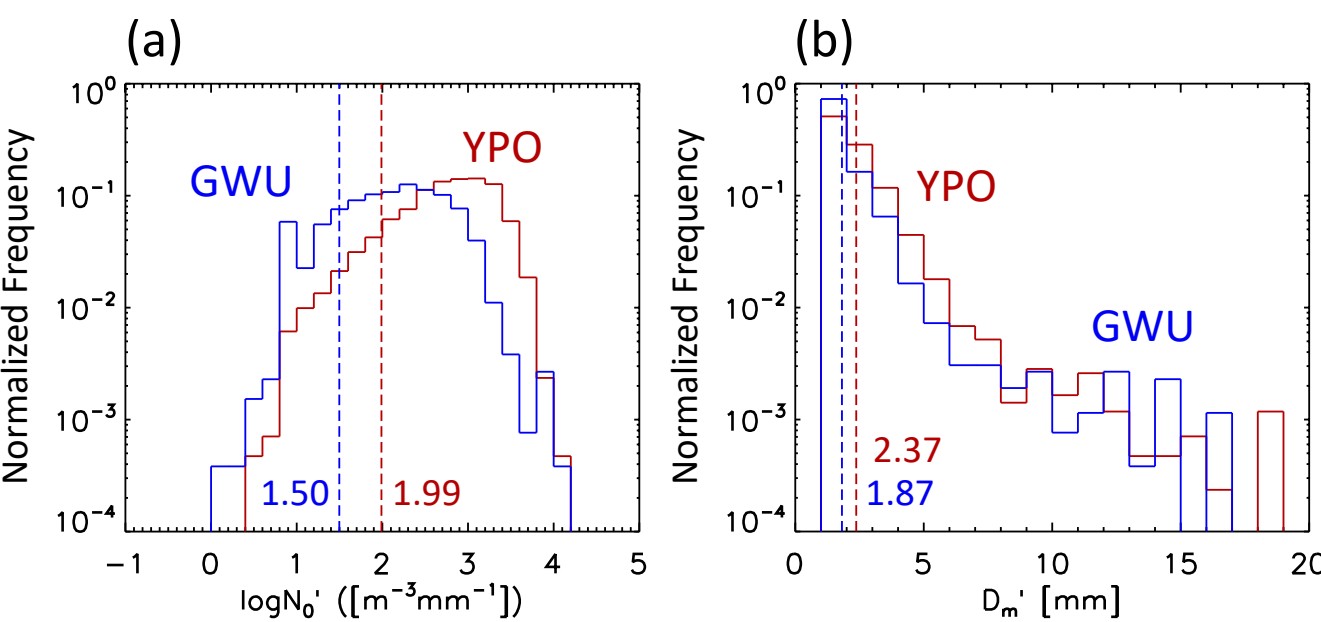

**Figure 4.** Normalized frequency of (a) $logN_0'$ and (b) $D_m'$ for YPO (in red) and GWU (in blue) sites. Dashed lines denote the mean value of each site.

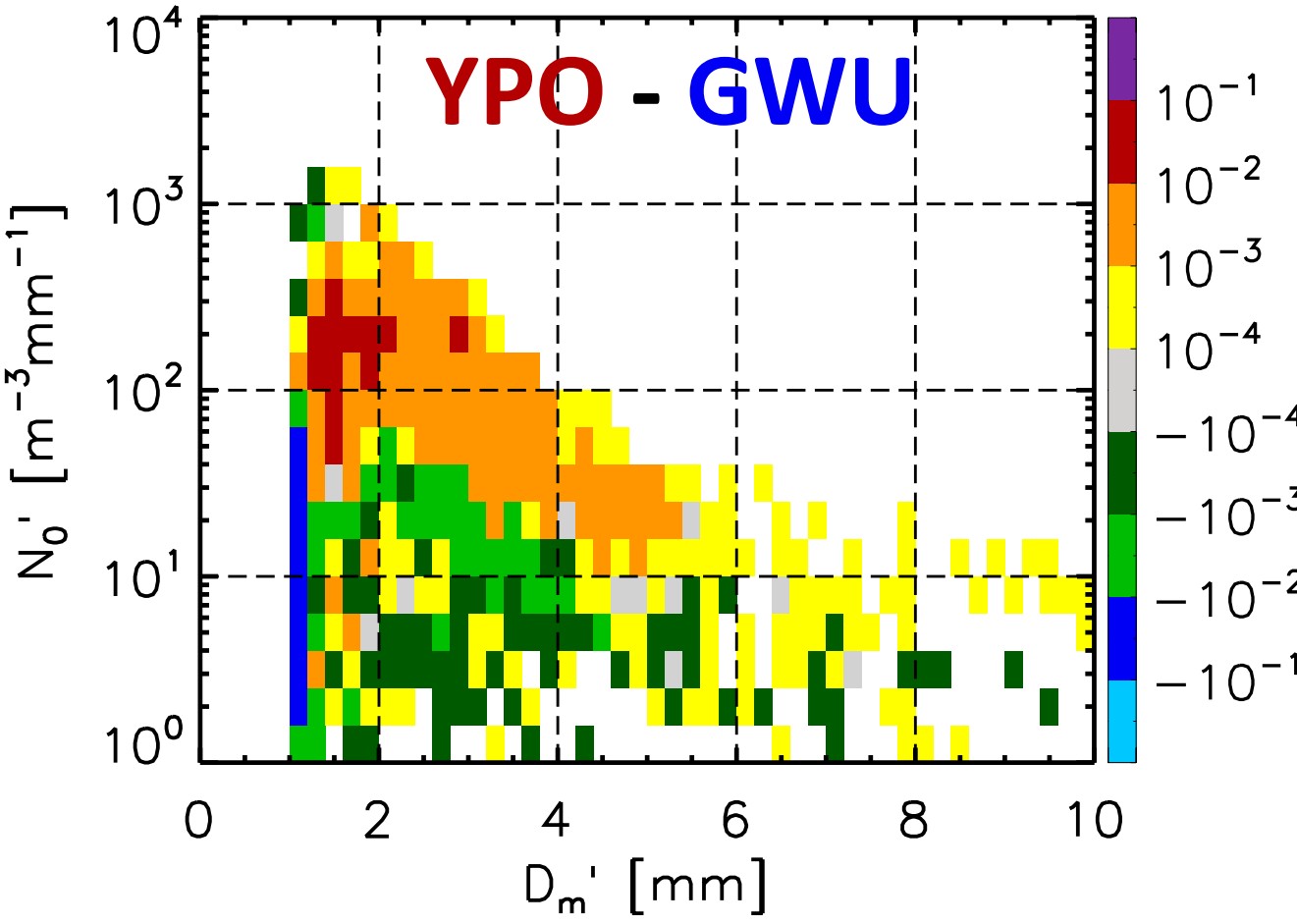

**Figure 5.** Difference of normalized frequency of $N_0'$–$D_m'$ between YPO and GWU sites. Warm (cold) color indicates YPO (GWU) has more frequency.



**Figure 6.** CFADs of radar reflectivity (dBZ, first row), negative of Doppler radial velocity ($\mathrm{m\,s^{-1}}$, second row), and spectral width ($\mathrm{m\,s^{-1}}$, third row) from MRR (first three rows) and normalized frequency of $N_0'-D_m'$ (fourth row) at YPO, CPO, and GWU sites for the air–sea interaction events during 2–yr winters. The number of snow–observed minutes at the ground by PARSIVEL is shown in the upper right corner of the diagram in the fourth row.



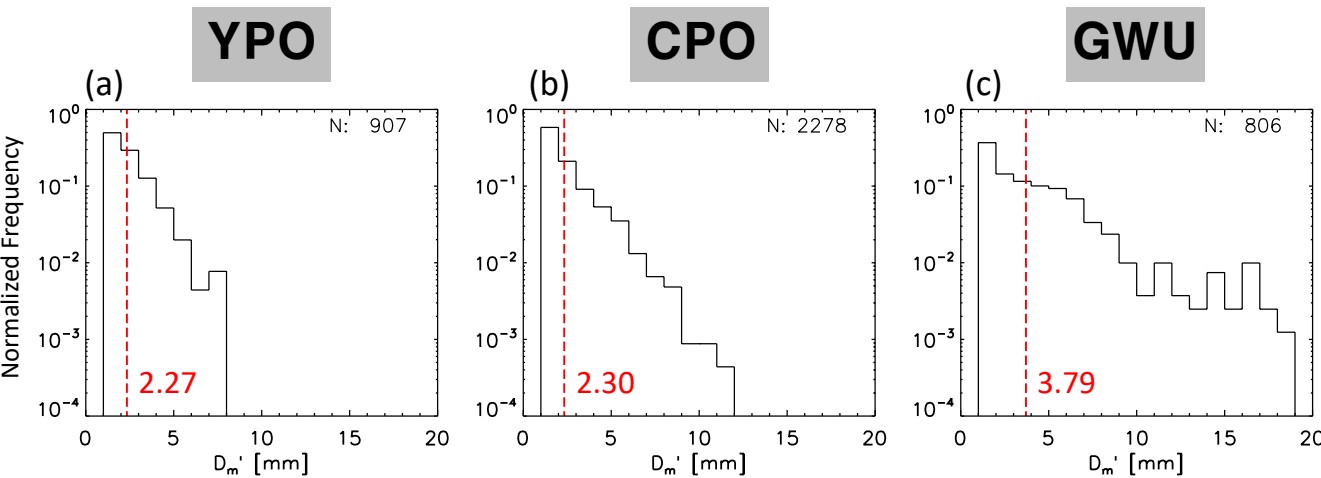

**Figure 7.** Normalized frequency of $D'_m$ at (a) YPO, (b) CPO, and (c) GWU sites for air–sea interaction events during 2–yr winters. Red dashed line indicates the average value for each diagram.



**Figure 8.** Same as in Fig. 6, except for the cold low events.

**Figure 9.** Normalized frequency of $D'_m$–$SR$ (first row) and $N'_0$–$SR$ (second row) at YPO and CPO sites for the cold low events.





**Figure 10.** Same as in Fig. 6, except for the warm low events.

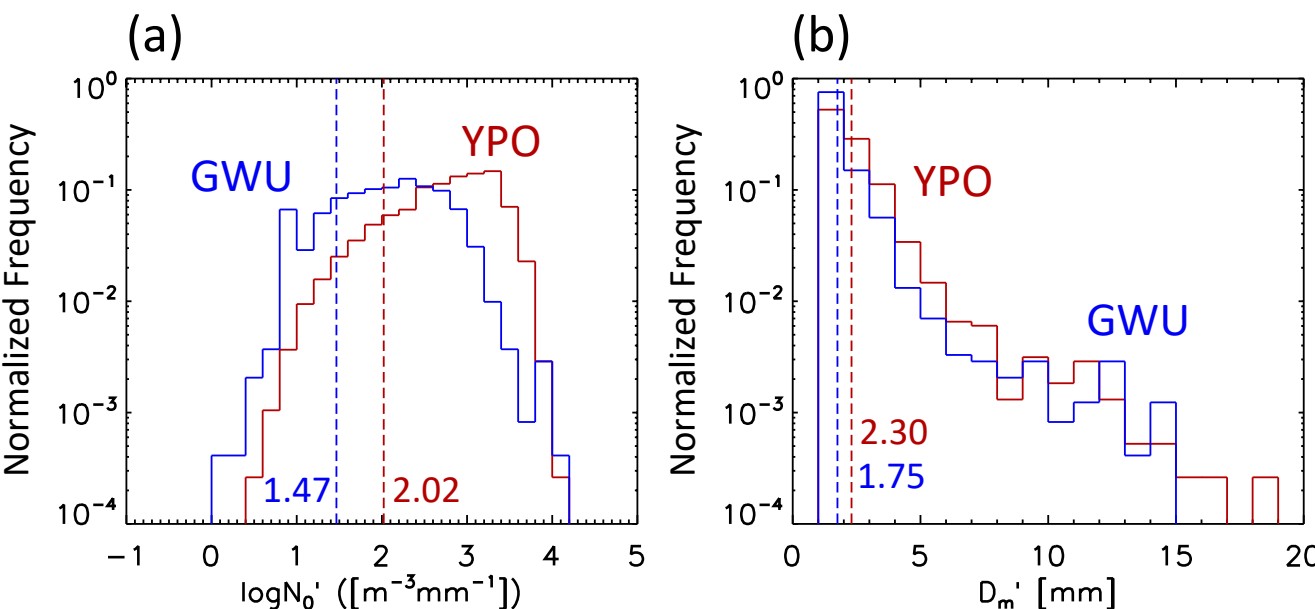

**Figure 11.** Same as in Fig. 4, except for the warm low events.