# Peer review of "Impact of wind pattern and complex topography on snow microphysics during ICE-POP 2018"

_Atmospheric Chemistry and Physics, 2021_

## Referee Comment (RC1)

**General Comments**

The authors are to be commended for a novel, scientifically useful, and high-quality study. I really enjoyed reading it. I would characterize my comments as relatively minor, but within these minor comments the most significant is the comment about the seeder-feeder process.

**Specific Comments**

L33-36: Consider adding a citation for a recent study on the effects of shear-induced turbulence on aggregation and riming: https://doi.org/10.1175/JAS-D-17-0365.1

L60-90: There are some recent research findings that should be mentioned somewhere in this portion of the paper. First, in sea-effect events, increasing the cross-barrier on-shore wind speed has been shown to increase the overall precipitation amounts, and to shift the precipitation farther inland and over the higher terrain (https://doi.org/10.1175/MWR-D-19-0007.1 and https://doi.org/10.1175/MWR-D-19-0390.1). Second, in situations when the Froude number is insufficient for the flow to move over a coastal mountain range, air can be blocked along the coast, shifting the sea-effect snowfall maximum into the low elevations along the coast (https://doi.org/10.1175/MWR-D-19-0390.1 and https://doi.org/10.2151/jmsj.2014-105).

L173-176: 80% rain seems like a very large fraction of rain for an algorithm to eliminate rain. Perhaps 40% would be more appropriate?

L180: Does the 'normalized frequency' include only precipitating periods? Is it frequency relative to all precipitating periods or frequency relative to all time elapsed during the 20 events?

Section 3: The information and analysis presented in this section is excellent.

L404-408: The data suggests to me that the increased turbulence in the 3–3.5 km layer is not just causing increased aggregation, but also riming. Below 3 km in Fig 3, the velocity spectrum shifts to the right (faster fall velocities). This would be consistent with graupel now appearing in the spectrum of hydrometeors.

L430-434: Aggregation becomes more likely at warmer temperatures. Does the airmass become colder (at a given altitude) as it moves inland? This may be beyond the scope of the paper – I am just curious.

L448-450: The authors have shown very convincing evidence that both riming and aggregation increase from YPO to CPO, but I think this sentence does not quite support the argument. In my opinion, the way to make the argument for both riming and aggregation would be to say something like "the doppler velocity spectrum at CPO has a similar median to that at YPO, but the spectrum is much wider, suggesting an increased frequency of both slow-falling aggregates and faster-falling rimed particles (Fig 8)." The next sentence, mentioning Figure 9, seems great and does not need any change.

L486-488: The difference in slope that the authors are speaking about here is very difficult to discern…perhaps consider reducing the range of the X-axis in the reflectivity CFADs and the slope will be more apparent?

L494-495: I'm confused by this statement, because it appears to me in Fig. 11b that GWU has a higher frequency than YPO in the smallest $D_m$ bin.

L484-504: I think that the authors have presented solid evidence for aggregation being the dominant growth mechanism at GWU, but I don't see how that indicates that the seeder-feeder process is happening. The seeder-feeder process involves a dual-layer cloud structure, and that has not been shown. This is the most significant issue with the paper, in my opinion.

L525: This paper (https://doi.org/10.1175/MWR2874.1) occurs in a somewhat comparable regime (sea-effect snowfall) and they document the tendency for increased riming over even relatively small peaks. It corroborates the results of the present study nicely.

**Technical Corrections**

Is it possible for the black-and-white figures be made into color figures? They are difficult to read.

Fig. 1: if the size of the markers were reduced by 30%, the reader could see the terrain in the vicinity of the sites a bit more easily.

L445: All I see in the PDF I am reading is "m s$^{-1}$" with no number. This could be an error by the Copernicus website.

L464: I can see "2.5 m s$^{-1}$", but the other wind speed in this line says "m s$^{-1}$" with no number. This could be an error by the Copernicus website.

Is it possible to list the mean liquid-precipitation-equivalent rate for each site in Figs 6, 8, and 10? It would be somewhat informative to know what these radar characteristics translate to as far as liquid rates.

---

## Author Comment (AC1)

**Impact of wind pattern and complex topography on snow microphysics during ICE-POP 2018**

**Author's Responses to Reviewers**

Kwonil Kim, Wonbae Bang, Eun-Chul Chang, Francisco J. Tapiador,
Chia-Lun Tsai, Eunsil Jung, and Gyuwon Lee

June 2021

We would like to thank the editor and reviewers for their careful reading of our manuscript and their constructive remarks. We sincerely appreciate all valuable comments and suggestions that helped us to improve the manuscript.

Please see below for our point-by-point response to all reviewers' comments. Our responses are highlighted with a box filled in gray, followed by the corresponding changes to the manuscript. Changes that have been made are marked in blue within the modified text represented by "...", while the removals are highlighted with the deleted red text (for example, red). The figure and table numbers in our responses are based on the revised manuscript.

We hope that our responses and the changes made to the manuscript will render it suitable for publication in Atmospheric Chemistry and Physics. In any case, we are open to consideration of any further comments on our answers.

**Reviewer 1**

**General Comments**

The authors are to be commended for a novel, scientifically useful, and high-quality study. I really enjoyed reading it. I would characterize my comments as relatively minor, but within these minor comments the most significant is the comment about the seeder-feeder process.

> We thank the referee for the careful and insightful review of our manuscript. We highly appreciate the positive assessment and constructive suggestions. A point-by-point response to the comment is given below, including the one on the seeder-feeder process.

**Specific Comments**

**1.1.** L33-36: Consider adding a citation for a recent study on the effects of shear-induced turbulence on aggregation and riming: https://doi.org/10.1175/JAS-D-17-0365.1.

> We have included a citation to the Barnes et al. (2018) paper.
>
> ---
>
> " Under strong shear conditions, the turbulent cell and induced updraft are favored, being responsible for the production of a considerable amount of supercooled water that promotes both aggregation and riming as they increase the probability of collision between hydrometeors, in particular, snowflakes and supercooled water droplets (Houze and Medina, 2005; Barnes et al. 2018)."

**1.2.** L60-90: There are some recent research findings that should be mentioned somewhere in this portion of the paper. First, in sea-effect events, increasing the cross-barrier on-shore wind speed has been shown to increase the overall precipitation amounts, and to shift the precipitation farther inland and over the higher terrain (https://doi.org/10.1175/MWR-D-19-0007.1 and https://doi.org/10.1175/MWR-D-19-0390.1). Second, in situations when the Froude number is insufficient for the flow to move over a coastal mountain range, air can be blocked along the coast, shifting the sea-effect snowfall maximum into the low elevations along the coast (https://doi.org/10.1175/MWR-D-19-0390.1 and https://doi.org/10.2151/jmsj.2014-105).

> Thanks for the reviewer's suggestions. We have referenced the suggested papers.
>
> ---
>
> " They reported that the Froude number, which can represent the blocking degree, is important to determine whether the precipitation amount is more in the coastal region or in the mountainous region. Insufficient Froude number favors an occurrence of convergence between blocked and environmental flow, and shifting maximum precipitation to the coastal area (Ohigashi et al. 2014; Veals et al. 2020). On the other hand, when Froude number is large enough, the maximum precipitation tends to occur over the mountainous or inland area (Veals et al. 2019; Veals et al. 2020)."
>
> ---
>
> " In relation to the Taebaek mountains, the Kor'easterlies can be interpreted as a cross-barrier flow. The higher cross-barrier wind speed tends to have stronger precipitation rate (Veals et al. 2019)."

**1.3.** L173-176: 80% rain seems like a very large fraction of rain for an algorithm to eliminate rain. Perhaps 40% would be more appropriate?

> The referee is correct. The threshold has been changed as suggested. The corresponding figures have been regenerated: Figures 5, 6, 7, 8, 9, 10, 11, and 12. This threshold change affected mostly to the number of data of GWU PARSIVEL due to a warmer environment (lower altitude). The changes of the results were minor and there have been no need to change any scientific logics to reach our conclusions. The only corrections needed was just to change a few numbers in the manuscript. Furthermore, we identified a few numerical typos in line 399–400 and amended them in this revised manuscript.
>
> ---
>
> " If the number of raindrops was greater than 40% of total number of particles, we considered the major precipitation type as rain and discarded this time interval."
>
> ---
>
> " Normalized frequency of $N_0'$ and $D_m'$ for the YPO and GWU sites shows the mountainous site (YPO) has greater number concentration (mean value of 97.72 m$^{-3}$ mm$^{-1}$) and a larger diameter (mean value of 2.38 mm) compared with the coastal site (GWU) having a mean number concentration of 38.02 m$^{-3}$ mm$^{-1}$ and a mean diameter of 2.04 mm, leading to higher snowfall rate at YPO."
>
> ---
>
> " The mean values of characteristic diameter increase from west (YPO, 2.28 mm) to east (GWU, 3.88 mm)."
>
> ---
>
> " The mean values of number concentration and diameter of YPO (GWU) are 104.71 (34.67) m$^{-3}$ mm$^{-1}$ and 2.30 (1.97) mm, respectively, indicating the mountainous region is characterized by heavier snowfall with more and larger snowflakes"

**1.4.** L180: Does the 'normalized frequency' include only precipitating periods? Is it frequency relative to all precipitating periods or frequency relative to all time elapsed during the 20 events?

> We have now clarified it in the text. The reason for the normalization by this value is to make CFADs from different sites to be comparable by dividing the same value.
>
> ---
>
> " Here, the frequency distribution is normalized by the number of all times elapsed during the events listed in Table 1."

**1.5.** Section 3: The information and analysis presented in this section is excellent.

> Many thanks for the positive evaluation.

**1.6.** L404-408: The data suggests to me that the increased turbulence in the 3–3.5 km layer is not just causing increased aggregation, but also riming. Below 3 km in Fig 3, the velocity spectrum shifts to the right (faster fall velocities). This would be consistent with graupel now appearing in the spectrum of hydrometeors.

> Amended.
>
> ---
>
> " This implies that higher snowfall rate is frequent  at YPO with larger snowflake due to more active aggregation as well as stronger riming."

**1.7.** L430-434: Aggregation becomes more likely at warmer temperatures. Does the airmass become colder (at a given altitude) as it moves inland? This may be beyond the scope of the paper – I am just curious.

> We agree with the reviewer that at a given height the warmer temperature is favorable condition for aggregation, however, in this study, the aggregated snow at the ground at GWU site was due to a warmer temperature of *lower altitudes*. Despite that, we have checked if the coastal sites were warmer than mountainous sites at a given height. The Fig. S1 shows the temperature profiles from upper-air soundings at three sites (DGW, BKC, and GWW) for the air–sea interaction event. Please see purple filled circles in Fig. 1 for the site location. The result suggests that the temperature of airmass did not decrease as the airmass moves to the mountainous region. The temperature at DGW was even about 1 °C higher than those of coastal sites (Fig. S1a), indicating the aggregation at GWU is likely contributed by the warmer temperature of lower altitudes.
>
>
[Figure]

>
> Figure S1: Temperature and wind profiles for (a) Mar 15 2018 18:00 UTC and (b) Mar 16 2018 00:00 UTC from radiosonde launched at DGW, BKC, and GWW sites.

**1.8.** L448-450: The authors have shown very convincing evidence that both riming and aggregation increase from YPO to CPO, but I think this sentence does not quite support the argument. In my opinion, the way to make the argument for both riming and aggregation would be to say something like "the doppler velocity spectrum at CPO has a similar median to that at YPO, but the spectrum is much wider, suggesting an increased frequency of both slow-falling aggregates and faster-falling rimed particles (Fig 8)." The next sentence, mentioning Figure 9, seems great and does not need any change.

> Revised. Thank you for the suggestion.
>
> ---
>
> " Comparing two sites,  the Doppler velocity spectrum at CPO has a similar median to that at YPO, but the spectrum is much wider, suggesting an increased frequency of both slow-falling aggregates and faster-falling rimed particles (Fig. 9). "

**1.9.** L486-488: The difference in slope that the authors are speaking about here is very difficult to discern... perhaps consider reducing the range of the X-axis in the reflectivity CFADs and the slope will be more apparent?

> The ranges of the x-axis in the reflectivity CFADs have been reduced (Figure 11). In addition, for the consistency, the x-axis ranges of reflectivity CFADs in Figures 4, 7, and 9 have been reduced. This was actually the reason why quantitative values have been provided in the sentence.

**1.10.** L494-495: I'm confused by this statement, because it appears to me in Fig. 11b that GWU has a higher frequency than YPO in the smallest Dm bin.

> Sorry for the confusion. It is correct that GWU has a higher frequency than YPO in the smallest $D'_m$ bin. In fact, our intention is not to argue that aggregation is *more* active at GWU *than YPO*, but GWU favors to have aggregated snow. The snow growth by aggregation is also important at YPO as explained at the last sentence of the same paragraph: "aggregation process is also an important growth mechanism in mountainous regions." Regarding the higher frequency in the smallest $D_m$ bin, most of the timesteps with small $D'_m$ in Fig. 12b are corresponding to the small $N'_0$ (Fig. 11l), indicating weak precipitation with small particles. Since our focus is the dominant snow growth mechanism that appears in the first peak of reflectivity frequency, we prefer to keep the sentence by adding a clarifying statement as follow:
>
> ---
>
> " The low frequency of high number concentration above $500 \text{ m}^{-3} \text{ mm}^{-1}$ for a small diameter ($< 3$ mm) in Fig. 11l confirms that small snowflakes are mostly consumed by aggregation dominantly, except for the case of weak precipitation with small $N'_0$ and small $D'_m$."

**1.11.** L484-504: I think that the authors have presented solid evidence for aggregation being the dominant growth mechanism at GWU, but I don't see how that indicates that the seeder-feeder process is happening. The seeder-feeder process involves a dual-layer cloud structure, and that has not been shown. This is the most significant issue with the paper, in my opinion.

> Fig. 3f actually shows a two-layered cloud structure. Based on the strong thermal inversion layer with the maximum directional wind shear, the lower cloud was formed by air–sea interaction whereas the upper cloud was associated with the frontal precipitation system.
> Besides, we have added a new figure in the revised manuscript (Figure 3) that shows two layers of clouds separated from each other and added the following paragraph:
>
> ---
>
> " As discussed above, a seeder–feeder process is expected due to "seeding" precipitation from frontal precipitation system and air–sea effect (and orographic) "feeder" cloud. A dual-layered cloud structure, which is necessary for seeder–feeder process, can be confirmed by looking at an example of typical vertical structure of warm low event from upper-air sounding (Fig. 3). A distinct lower cloud pre-existed below 800 hPa with the Kor'easterlies at 09:00 UTC (Fig. 3a), and an upper layer (associated with westerly) become saturated from high altitude (300 hPa at 09:00 UTC) to middle layer (540 hPa at 12:00 UTC) (Fig. 3a–b). Then all the altitudes are covered by the clouds with an apparent thermal inversion layer collocated with the transition zone between easterly below and westerly aloft (Fig. 3c). These three plots indicate the upper cloud and lower cloud were separated initially then combined later as the low pressure center moves closer to the south of the studied area. Note that this kind of vertical structure was evident in warm low events."
>
> ---
>
> Another evidence of the dual-layered structure can be found from the time–height plot of the vertically pointing X-band radar measurement (Fig. S2). While the lower cloud (associated with the system developed by air–sea interaction) started to appear from 09:00 UTC where the easterly/southeasterly was dominant, the upper cloud (associated with frontal precipitation system) appeared at which the westerly predominant. The depth of lower cloud was increased

until the system became indistinguishable from the upper cloud at around 12:00 UTC.

[Figure]

Figure S2: Time–height plots from Mar 04 2018 event for reflectivity from vertically pointing X-band radar at MHS. The wind profiles from the upper-air sounding at DGW (mountainous area) are overlayed.

In addition, reading the referee's comment, we think the original manuscript can potentially mislead the readers. For clarification, the paragraph has been divided in this version since we did not mean that the fact that aggregation is dominant growth mechanism at GWU indicates the seeder-feeder process is happening.
* * *
" A possible evidence of snowfall enhancement by seeder–feeder mechanism (Bergeron, 1965) may be found in the continuous increase in reflectivity of the primary peak (see normalized frequency > 2.5%).
Comparing mountainous regions (YPO and CPO) and coastal regions (GWU), we can see that the increase of reflectivity at the GWU site below 3 km ... "

**1.12.** L525: This paper (https://doi.org/10.1175/MWR2874.1) occurs in a somewhat comparable regime (sea-effect snowfall) and they document the tendency for increased riming over even relatively small peaks. It corroborates the results of the present study nicely.

According to the reviewer's suggestion, we added the Kusunoki et al. (2005) paper in the revised manuscript.
* * *
" MRR observations for air-sea interaction suggests that riming exists at all sites. This may be explained by low-level updraft induced by orographic lifting and convective instability over ocean due to the Kor'easterlies. This can be supported by the observational evidence presented by Kusunoki et al. (2005) that terrain-induced updraft favors the production of supercooled liquid water which leads to snow growth by riming, even over the small-scale (horizontal scale of approximately 3 km) terrain. Updraft is more frequent in mountainous sites, leading to a stronger signature of riming."

**Technical Corrections**

**1.13.** Is it possible for the black-and-white figures be made into color figures? They are difficult to read.

> Revised in Figures 7, 9, 10, and 11 accordingly.

**1.14.** Fig. 1: if the size of the markers were reduced by 30%, the reader could see the terrain in the vicinity of the sites a bit more easily.

> Corrected as suggested.

**1.15.** L445: All I see in the PDF I am reading is "$\mathrm{m\,s^{-1}}$" with no number. This could be an error by the Copernicus website.

> Corrected.
>
> ---
>
> " Note that the median wind speed of all cold low events was 6.70 $\mathrm{m\,s^{-1}}$ at DGW located between YPO and CPO, while they were 2.60 and 2.30 $\mathrm{m\,s^{-1}}$ for air-sea interaction and warm low events, respectively."

**1.16.** L464: I can see "2.5 $\mathrm{m\,s^{-1}}$", but the other wind speed in this line says "$\mathrm{m\,s^{-1}}$" with no number. This could be an error by the Copernicus website.

> Corrected.
>
> ---
>
> " Both the frequency of Doppler velocity above 1.5 $\mathrm{m\,s^{-1}}$ and primary frequency peak of Doppler velocity of an approximately 1.0 $\mathrm{m\,s^{-1}}$ rise in that layer $(3.0 - 3.5$ km) and maintain their velocity as they fall."

**1.17.** Is it possible to list the mean liquid-precipitation-equivalent rate for each site in Figs 6, 8, and 10? It would be somewhat informative to know what these radar characteristics translate to as far as liquid rates.

> Good point. Unfortunately, it was not able to get the liquid-equivalent precipitation rate measurements from collocated weighing gauges for the entire period of the studied events. For instance, the Pluvio weighing gauge at YPO was deployed from March 2017, meaning the observations of at least five events of the selected events (Table 1) in January and February are not available. Moreover, no weighing gauge was operated at GWU in 2016-2017 winter season. Instead, in response to this comment, we have calculated the mean liquid-precipitation-equivalent rate at 1.5 km height (for each site) from the $Z_e$-S relationship obtained at the MRR wavelength (Souverijns et al. 2017). They are now listed in the captions of Figures 4, 7, 9, and 11.

**Reviewer 2**

**General Comments**

The manuscript documents the microphysical characteristics of snow by analyzing the PARSIVEL and MRR data collected from ICE-POP 2017-2018. The snow events were classified by three different synoptic systems. The three type of synoptic systems are air-sea, cold low and warm low patterns. The results show distinct characteristics of snow. The aggregation process increases the size of the snow. The riming process has higher values of fall velocity of snow particle. Most of the conclusions are reasonable, but further detailed analysis is missing. However, the manuscript did a good job summarizing 20 snow events from ICE-POP 2017-2018.

> We would like to thank the reviewer for careful reading of this manuscript and the positive review. While we are sorry to hear the indefinite comment "further detailed analysis is missing", we believe that we have proposed a new understanding on the impact of wind flow and topography on the microphysics in this region following the objective: "The purpose of this paper is to elucidate the microphysical characteristics of snow in the Gangwon region in both different heights and surface by using MRR and PARSIVEL datasets." Please note that the future study will cover more detailed view of microphysics and dynamics for each synoptic system by means of high-resolution radar-based products (e.g., 3D wind field) and polarimetric radar variables. A point-by-point response to the comment is given below. We hope the referee find our responses and the revised manuscript satisfactory.

**Minor comments**

**2.1.** Line 55: "showed that the mean precipitation amount increased by about 45% in the presence of both Kaema high and low pressure. " Any reference?

> Song et al. (2016) showed that the coexistence of the Kaema high and low pressure increases the mean precipitation amount by about 45%. From the pronoun "They" in the original manuscript, we believe the readers can easily recognize the reference from the manuscript: "Song et al. (2016) classified the synoptic environments.... *They* showed that the mean precipitation amount....." Thus, we have decided to keep the sentences in the current form without citing the same article additionally.

**2.2.** Line 419: "the size of the snow does not greatly enlarge." Why? Evidence?

> It is obvious that the increase of the dimension during riming is not significant compared to aggregation process. While the size (or volume) does not increase much, the mass increases during riming, leading to increases of density and fall speed of snowflake, and the Doppler velocity. This statement is partially based on a conceptual model of the growth of snow by riming of Heymsfield et al. (1982) which has been adopted in the current microphysical parameterizations of riming (Morrison and Grabowski 2008; Jensen and Harrington, 2015; Morrison and Milbrandt, 2015). Based on the model, during riming, it is hypothesized that the rime is accreted to the interstices between the crystal branches while the maximum dimension is conserved until it becomes a spherical shape. Although a new parameterization was suggested recently because the increase of maximum dimension during riming was found from the simulation (Seifert et al. 2019), the increase should be limited to the graupel size of typically 2–5 mm (Heymsfield et al. 2018). The citation is added in the manuscript:

> " When riming occurs, Doppler velocity is increased because the mass of snow increases due to collection of supercooled liquid water, while the size of the snow does not greatly enlarge (Heymsfield et al. 1982), resulting in nearly constant or smaller increase of reflectivity. "

**2.3.** Line 422: Where is the second peak of reflectivity?

> Clarified. The second peak of reflectivity can be found at the lower reflectivity below 3 km height, while the primary peak appears at the reflectivity greater than 15 dBZ.
>
> " In the layer where the updraft is present, there is a secondary peak of reflectivity (at lower reflectivity less than 12 dBZ) and enhanced spectral width in mountainous sites up to a height of 3 km. "

**References**

Barnes, H. C., J. P. Zagrodnik, L. A. McMurdie, A. K. Rowe, and R. A. Houze, 2018: Kelvin-Helmholtz waves in precipitating midlatitude cyclones. J. Atmos. Sci., 75, 2763–2785, https://doi.org/10.1175/JAS-D-17-0365.1.

Heymsfield, A., 1982: A Comparative Study of the Rates of Development of Potential Graupel and Hail Embryos in High Plains Storms. J. Atmos. Sci., 39, 2867–2897, https://doi.org/10.1175/1520-0469(1982)039<2867:ACSOTR>2.0.CO;2.

Heymsfield, A., M. Szakáll, A. Jost, I. Giammanco, and R. Wright, 2018: A comprehensive observational study of graupel and hail terminal velocity, mass flux, and kinetic energy. J. Atmos. Sci., 75, 3861–3885, https://doi.org/10.1175/JAS-D-18-0035.1.

Jensen, A. A., and J. Y. Harrington, 2015: Modeling Ice Crystal Aspect Ratio Evolution during Riming: A Single-Particle Growth Model. J. Atmos. Sci., 72, 2569–2590, https://doi.org/10.1175/JAS-D-14-0297.1.

Kusunoki, K., and Coauthors, 2005: Observations of quasi-stationary and shallow orographic snow clouds: Spatial distributions of supercooled liquid water and snow particles. Mon. Weather Rev., 133, 743–751, https://doi.org/10.1175/MWR2874.1.

Morrison, H., and W. W. Grabowski, 2008: A novel approach for representing ice microphysics in models: Description and tests using a kinematic framework. J. Atmos. Sci., 65, 1528–1548, https://doi.org/10.1175/2007JAS2491.1.

Morrison, H., and J. A. Milbrandt, 2015: Parameterization of cloud microphysics based on the prediction of bulk ice particle properties. Part I: Scheme description and idealized tests. J. Atmos. Sci., 72, 287–311, https://doi.org/10.1175/JAS-D-14-0065.1.

Ohigashi, T., K. Tsuboki, Y. Shusse, and H. Uyeda, 2014: An intensification process of a winter broad cloud band on a flank of the mountain region along the Japan-Sea coast. J. Meteorol. Soc. Japan, 92, 71–93, https://doi.org/10.2151/jmsj.2014-105.

Seifert, A., J. Leinonen, C. Siewert, and S. Kneifel, 2019: The Geometry of Rimed Aggregate Snowflakes: A Modeling Study. J. Adv. Model. Earth Syst., 11, 712–731, https://doi.org/10.1029/2018MS001519.

Song, J.-A., J. G. Lee, and Y.-J. Kim, 2016: The Study of Correlations between Air-Sea Temperature Difference and Precipitation and between Wind and Precipitation in the Yeongdong Coastal Region in Relation to the Siberian High. Atmosphere, 26, 127–140, https://doi.org/10.14191/Atmos.2016.26.1.127.

Souverijns, N., and Coauthors, 2017: Estimating radar reflectivity - Snowfall rate relationships and their uncertainties over Antarctica by combining disdrometer and radar observations. Atmos. Res., 196, 211–223, https://doi.org/10.1016/j.atmosres.2017.06.001.

Veals, P. G., W. J. Steenburgh, S. Nakai, and S. Yamaguchi, 2019: Factors affecting the inland and orographic enhancement of sea-effect snowfall in the hokuriku region of Japan. Mon. Weather Rev., 147, 3121–3143, https://doi.org/10.1175/MWR-D-19-0007.1.

Veals, P. G., W. J. Steenburgh, S. Nakai, and S. Yamaguchi, 2020: Intrastorm Variability of the Inland and Orographic Enhancement of a Sea-Effect Snowstorm in the Hokuriku Region of Japan. Mon. Weather Rev., 148, 2527–2548, https://doi.org/10.1175/MWR-D-19-0390.1.